# Sesamin’s Therapeutic Actions on Cyclophosphamide-Induced Hepatotoxicity, Molecular Mechanisms, and Histopathological Characteristics

**DOI:** 10.3390/biomedicines11123238

**Published:** 2023-12-07

**Authors:** Abdulmajeed M. Jali, Mohammad Firoz Alam, Ali Hanbashi, Wedad Mawkili, Basher M. Abdlasaed, Saeed Alshahrani, Abdullah M. Qahl, Ahmad S. S. Alrashah, Hamad Al Shahi

**Affiliations:** 1Department of Pharmacology and Toxicology, College of Pharmacy, Jazan University, Jazan 45142, Saudi Arabia; amjali@jazanu.edu.sa (A.M.J.); ahanbashi@jazanu.edu.sa (A.H.); wmawkili@jazsnu.edu.sa (W.M.); saalshahrani@jazanu.edu.sa (S.A.); phd.qahl@gmail.com (A.M.Q.); ahmadss-r@hotmail.com (A.S.S.A.); halshahi@jazanu.edu.sa (H.A.S.); 2Department of Pharmacology, University of Oxford, Mansfield Road, Oxford OX1 3QT, UK; 3Department of Biology, Faculty of Education, Alasmaray Islamic University, Zliten 218521, Libya; edu@asmarya.edu.ly; 4Pharmacy Department, Jazan University Hospital, Jazan University, Jazan 45142, Saudi Arabia; 5Pharmacy Administration, Ministry of Health, Health Affairs General Directorate, Najran 66251, Saudi Arabia

**Keywords:** sesamin, cyclophosphamide, hepatoprotective, hepatotoxicity, histopathology, cytokines, apoptosis

## Abstract

Cyclophosphamide, an alkylating agent integral to specific cancer chemotherapy protocols, is often curtailed in application owing to its significant hepatotoxic side effects. Therefore, this study was conducted to assess the hepatoprotective potential of sesamin, a plant-originated antioxidant, using rat models. The rats were divided into five groups: a control group received only the vehicle for six days; a cyclophosphamide group received an intraperitoneal (i.p.) single injection of cyclophosphamide (150 mg/kg) on day four; a sesamin group received a daily high oral dose (20 mg/kg) of sesamin for six days; and two groups were pretreated with oral sesamin (10 and 20 mg/kg daily from day one to day six) followed by an i.p. injection of cyclophosphamide on day four. The final and last sesamin dose was administered 24 h before euthanasia. At the end of the experiment, blood and liver tissue were collected for biochemical and histopathological assessments. The results indicated significantly increased liver markers (AST, ALT, ALP, and BIL), cytokines (TNFα and IL-1β), caspase-3, and malondialdehyde (MDA) in the cyclophosphamide group as compared to the normal control. Additionally, there was a significant decline in antioxidants (GSH) and antioxidant enzymes (CAT and SOD), but the sesamin treatment reduced liver marker enzymes, cytokines, and caspase-3 and improved antioxidants and antioxidant enzymes. Thus, sesamin effectively countered these alterations and helped to normalize the histopathological alterations. In conclusion, sesamin demonstrated the potential for attenuating cyclophosphamide-induced hepatotoxicity by modulating cytokine networks, apoptotic pathways, and oxidative stress, suggesting its potential role as an adjunct in chemotherapy to reduce hepatotoxicity.

## 1. Introduction

The use of modern anticancer drugs has led to significant improvements in patient outcomes and survival rates; nevertheless, these drugs have also been linked to a number of serious adverse effects, such as neurotoxicity, nephrotoxicity, cardiotoxicity, and hepatotoxicity [1,2]. The liver is highly vulnerable to damage generated by medications, including cytotoxic chemotherapy regimens, due to its central involvement in the breakdown of pharmaceuticals and poisons. In addition, medication-induced liver injury can have a cumulative impact, whereby ongoing damage feeds back to cause problems with drug metabolism and increased toxicity. Damage to the liver’s sinusoids, blood vessels, bile ducts, or hepatocytes can all lead to hepatotoxicity. Also, damage to the liver can be caused by the obstruction of blood vessels and bile ducts, the production of toxic metabolites, and the infiltration of inflammatory cells into the liver parenchyma. These negative consequences are problematic. Thus, close collaboration between oncologists and hepatologists to see the patient’s liver condition and medical interference to prevent lasting hepatic injury are needed as new targeted cancer medicines are developed [3,4]. Therefore, when hepatotoxicity occurs, implementing methods such as stopping or reducing the dose of the pharmacological agent or adding a complementary medicine can decrease hepatotoxicity and enhance therapy outcomes [5].

Since known chemotherapies’ hepatotoxicity have been extensively discussed, this study will just briefly touch on the associated liver toxicities and their protection. Antitumor antibiotics, antimetabolites, alkylating drugs, platinum agents, antimicrotubular agents, and topoisomerase inhibitors are among the classes of chemotherapeutics that cause hepatotoxicity. Historically, broad-spectrum chemotherapeutics have served as the foundation of anticancer treatment plans. Nevertheless, they frequently cause cellular damage that impairs cell division or causes cells that divide quickly to become apoptotic. This can be attributed to traditional chemotherapeutics’ nonselective properties that lead to negative adverse effects and cause hepatotoxicity. Elevated liver markers in blood and hepatic disorders are just a few examples of the hepatotoxic side effects of conventional chemotherapy [6,7,8]. The current practice for treating chemotherapy-induced hepatotoxicity has been to reduce pharmaceutical doses in the presence of hepatotoxicity or to stop using the offending mediator if the hepatic issue is resistant to dose declines. Thus far, new cases of hepatotoxicity are still being reported. Hence, traditional chemotherapeutics are still being investigated to better understand the molecular mechanism of hepatotoxicity and prevent associated side effects. 

The mechanism of methotrexate-related hepatotoxicity revealed that the production of reactive oxygen species (ROS), elevation of pro-apoptotic mediators, and activation of inflammatory pathways and cytokines contribute to liver damage. Interestingly, 18-glycyrrhetinic acid, a natural constituent of licorice, attenuated the hepatotoxic effects of methotrexate by decreasing the production of the mentioned proteins [9]. Additionally, a study examined the hepatotoxicity mechanisms of doxorubicin that increased levels of pro-apoptotic genes, including Bax, ROS generation, membrane lipid peroxidation, and altered mitochondrial energy metabolism all contributed to liver damage [10]. Hence, new treatments to lessen these harmful hepatic consequences can be measured and developed as counteracts to the hepatotoxicity mechanism of chemotherapy. Recently, efforts have been made to decrease the negative impacts of chemotherapeutic medications by increasing the use of natural antioxidants. Despite this shift, few experimental studies have explored ways to lessen the effects of cyclophosphamide on the liver. The literature reveals that some well-known natural antioxidants, like sesamin, are on hand; nevertheless, the vast majority of these compounds have not been studied.

Sesamin, a prominent lignin that originates in sesame seed oil (*Sesamum indicum*), is efficiently absorbed and distributed within the body. It has numerous metabolites that are widely distributed in the body’s organs. It has a number of physiological effects, including antioxidant [11], anti-inflammatory [12], antimitochondrial toxicity [13], anti-lipid profile and blood pressure [14], anti-alcohol-mediated hepatic injury [15], and anti-allergic effect [16]. In a prior research investigation, it was elucidated that sesamin augments the protective defense mediated by nuclear factor erythroid 2-related factor 2 (Nrf2) against oxidative stress and inflammation in colitis. This enhancement is facilitated through the activation of protein kinase B (AKT) and extra signal-regulated kinases (ERK) [17]. However, to our knowledge, sesamin’s protective effects against cyclophosphamide-induced hepatotoxicity have not been investigated. Having variable sources of natural compounds that can be effective in treating or preventing toxic effects induced by chemotherapeutics is highly motivating. This could aid in enhancing chemotherapy outcomes and improving quality of life. Therefore, the purpose of this study is to examine sesamin’s possible protective effects and explore its protective mechanisms against cyclophosphamide-induced hepatotoxicity.

## 2. Materials and Methods

### 2.1. Chemicals and Drug

Sigma Aldrich Co., St. Louis, MO, USA, supplied sesamin, thiobarbituric acid reactive substances, 5,5 -dithiobis (2-nitrobenzoic acid), hydrogen peroxide, reduced glutathione, trichloroacetic acid, and sulfosalicylic acid. The assay kits for AST, bilirubin, ALP, and ALT were obtained from Randox Lab Ltd., Antrim, UK, while IL-1β, TNFα, and caspase-3 were obtained from Abcam, Cambridge, UK. The assays were performed according to the manufacturer’s procedure.

### 2.2. Animal Models and Experimental Design

The Medical Research Centre at Jazan University, SA, supplied male Wistar rats (150–160 g). Before the experiment, these rats were familiarized with standard laboratory conditions, which encompassed a 12 h light-dark cycle, an ambient temperature of 20 ± 2 °C, and a relative humidity of 50 ± 15%. The study received ethical clearance with the reference REC41/1-032B.

The rats were divided into five groups: a control group received only the vehicle (saline solution) for six days; a cyclophosphamide group received an intraperitoneal (i.p.) single injection of cyclophosphamide (150 mg/kg) on day four [18]; a sesamin group received a daily high oral dose (20 mg/kg) of sesamin for six days; and two groups were pretreated with oral sesamin (10 and 20 mg/kg daily from day one to day six) followed by an i.p. injection of cyclophosphamide (150 mg/kg) on day four [19].

Following a treatment period of six days, specifically on the seventh day, the rats had a fasting period lasting no less than 12 h. Each group of rats was administered chloral hydrate anesthesia, following which blood samples were promptly obtained from them via the tail vein. Subsequently, the rat underwent sacrificial procedures, during which the liver extracted from each rat was weighed and subsequently partitioned into two distinct portions. The initial components were retained at a temperature of −80 °C to facilitate tissue preparation and biochemical analysis. On the other hand, some portion of the liver was submerged in a formalin solution (10%) for the purpose of conducting a histological examination. The blood samples were centrifuged for 15 min at 300 revolutions per minute (rpm). The resulting serum was then extracted and stored at −20 °C for the purpose of conducting liver function tests. The tissue sample was subjected to homogenization using a 10% solution of 0.1 M phosphate-buffered solution (PBS) and with a pH of 7.4 specifically prepared for use with liver samples. Homogenized tissue was centrifuged at 3000 rpm for 15 min at 4 °C. The resulting upper layer was utilized for the purpose of estimating oxidative stress and conducting a cytokine assay.

### 2.3. Biochemical Assay for Liver Functions

The standard method of the RandoxTM assay kit was used to assay the serum samples for aspartate transaminase, alanine transaminase, and bilirubin.

### 2.4. Inflammatory Cytokines and Apoptosis Marker Test

The estimation of IL-1β, TNFα, and caspase 3 was carried out by an ELISA assay technique. These markers were estimated by the guide line of the manufacturer assay kit (Abcam, UK) with a Bio-Tek ELISA reader (ELX800, Vernon Hills, IL, USA) 450 nm (IL-1β and TNFα) and 405nm (Caspase-3). The manufacturer’s methodology was used to analyze the data.

### 2.5. Oxidative Stress Assay for Liver Tissue

The lipid peroxidation marker, malondialdehyde (MDA), was analyzed by Utley et al. (1967). Briefly, using a Shimadzu UV-1601 spectrophotometer made in Japan, the MDA concentration was gauged at a 535 nm wavelength. The results were then translated to nmol of TBARS produced per hour for each mg of protein utilizing a molar extinction coefficient (MEC) of 1.56 × 105 M^−1^ cm^−1^ [20].

The concentration of glutathione (GSH) was determined as per the procedure set by Jallow (1974). Succinctly, GSH’s absorbance was noted at 412 nm, and the derived value was represented in relation to the DTNB conjugate amount produced per mg of protein, using a coefficient of 13.6 × 103 M^−1^ cm^−1^ [21].

Catalase (CAT) levels were assessed following the method by Claiborne (1985). The CAT absorbance was measured at 240 nm. The outcomes were then expressed in relation to the quantity of H_2_O_2_ consumed every minute for each mg of protein, using an MEC of 43.6 × 10^3^ M^−1^ cm^−1^. This value was further denoted as nmole (−) epinephrine preserved with a MEC of 4.02 × 103 M^−1^ cm^−1^ [22].

The superoxide dismutase (SOD) was determined based on the procedure by Stevens et al. (2000). The quantification of SOD was performed by observing the auto-oxidation rate of (−) epinephrine at 480 nm [23].

### 2.6. Histopathological Assessment

In accordance with standard histology practices, the liver tissue was fixed in 10% formaldehyde before being embedded in paraffin [24]. These implanted samples were sliced to 5 mm thickness by microtome, and they were then fixed on a slide and stained with hematoxylin and eosin (H&E). The stained slide was next examined using a Leica microscope at a magnification of 40×. On a scale of 0 to 4, the indicators of swelling, necrosis, and restoration were graded. Briefly, the degree of liver necrosis was graded as follows: grade 0 indicated no changes in histology; grade 1 indicated hepatic necrosis near the central vein; grade 2 indicated liver necrosis in two zones; grade 3 indicated more than two necrosis around the central vein; and grade 4 indicated massive necrosis. Thus, the grading was assessed as follows: 0, 1, 2, 3, and 4 for normal, minor, mild, moderate, and severe injuries, respectively. Similar to this, the frequency of hepatocyte sinusoidal obstruction found in the hepatocyte region was used to rate liver regeneration on a scale of 0 to 4. A score of 0 indicated no sinusoidal obstruction, a score of 1 indicated one or two figures, a score of 2 indicated three to four figures, a score of 3 indicated five to six figures, and a score of 4 indicated more than seven figures. [25].

### 2.7. Protein Evaluation

Protein estimation was carried out in liver tissue by the Lowery et al. (1951) procedure using bovine serum albumin as a reference standard [26].

### 2.8. Statistical Analysis

Statistical software (GraphPad Prism 9) was used to analyze the data obtained from the various studies, and the value was represented as the mean value and standard deviation with six replicates. The Tukey–Kramer test was employed after ANOVA analysis to differentiate statistically significant values. Differences across groups were judged significant if *p* < 0.05.

## 3. Results

### 3.1. Effects of Sesamin on Liver Weight and Liver Function Tests

The results revealed that rat liver weight increased in the cyclophosphamide-treated group as compared to the control group. It was also noticed that in treatment with higher doses of sesamin, rat liver weight was significantly decreased while no significant changes were seen in lower doses of sesamin (Figure 1).

As seen in Figure 2, the cyclophosphamide group had significantly higher levels of AST, ALT, ALP, and BIL than the control group. These markers dropped dramatically after treatment with sesamin in both the sesamin 10 + cyclophosphamide and sesamin 20 + cyclophosphamide groups. Results show that when comparing the sesamin-only (20) group to the control group, no significant difference was found with a *p* > 0.05.

### 3.2. Effects of Sesamin on MDA and Antioxidant Enzymes

While MDA levels increased, GSH levels decreased in the cyclophosphamide group (Figure 3). On the other hand, the administration of sesamin in both doses of 10 and 20 mg showed a significant protective effect in reducing liver toxicity. The GSH level increased following sesamin treatments, and the MDA level returned to normal. Similar to GSH, CAT and SOD were also reduced after cyclophosphamide and recovered following sesamin treatment at both doses. Sesamin at a dose of 20 mg showed a greater decline in MDA content and a greater rise in GSH, CAT, and SOD content than at a 10 mg dose. There were no statistically significant differences between the control group and the sesamin-only group (*p* > 0.05).

### 3.3. Effects of Sesamin on Cytokines and Apoptotic Markers

The levels of inflammatory cytokines, specifically IL-1β, exhibited a significant rise in the liver following the administration of cyclophosphamide. This rise, however, was drastically reduced at the greater dose of sesamin (*p* < 0.001) than the lower dose. Cyclophosphamide significantly (*p* < 0.0001) enhanced tumor necrosis factor (TNFα) concentration in comparison to the control. However, oral sesamin supplementation considerably reduced TNFα as shown in the sesamin 10 + cyclophosphamide and sesamin 20 + cyclophosphamide groups compared to the cyclophosphamide group (Figure 4). Similarly, the concentration of caspase-3, an apoptotic marker, was significantly elevated after cyclophosphamide treatment compared to the control (*p* < 0.0001). Nevertheless, this increase in caspase-3 was then significantly decreased following oral intake of sesamin at both doses (*p* < 0.001). Sesamin in larger doses produced a more significant reduction in these markers than in lower doses. The sesamin-only group and the control showed no appreciable changes.

### 3.4. Effects of Sesamin on Liver Architecture

Histological alterations such as central areas of intense clotting, intense deterioration, severe inflammation, sinusoidal obstruction, and apoptosis were visible in the cyclophosphamide group (Figure 5B). As shown, the injury score was substantially high (score = 4). As demonstrated in Figure 5C, the liver’s cells significantly regenerate after prior treatment with 10 mg/kg of sesamin, and the liver damage score was only modest (score = 2). The histological structure was greatly improved during pretreatment with the highest dose (sesamin 20), and the lowest injury score (score = 1), as shown in Figure 5D. On the other hand, as can be seen in Figure 5A, the normal control section of the stained liver showed no signs of damage and a normal organization of hepatocytes. Likewise, no major structural alterations were seen in the liver tissue after pretreatment with the maximum dose of sesamin 20 alone, as shown in Figure 5E, which had an injury score of 0.

## 4. Discussion

A chemotherapeutic medication such as cyclophosphamide increases cancer patients’ survival rates. However, its hepatotoxicity cannot be disregarded due to its constant difficulties and consequences. It was reported that higher exposure to cyclophosphamide’s toxic metabolites resulted in an increased risk of liver toxicity. Therefore, this study’s main goal was to investigate how the metabolism of cyclophosphamide damages the liver and evaluate how sesamin reverses this toxicity. Liver enzymes cause cyclophosphamide to undergo metabolic modifications that result in 4-dyroxychlophosphamide. It further breaks down into two harmful metabolites, acrolein and phosphoramide mustard, which is the most toxic [24,25,26,27].

Idiosyncratic hepatotoxicity is caused by an unknown origin. In contrast, sinusoidal obstruction syndrome is most likely caused by the drug’s direct toxic effects on the liver’s sinusoidal cells, which lead to their necrosis and discharge into the sinusoids, obliterating hepatic veins, and creating obstruction. The liver extensively metabolizes cyclophosphamide into several metabolites of which over 150 have been identified. The major metabolites include hydroxypropyl-phosphoramide mustard, deschloroethyl-cyclophosphamide, 4-hydroxy-cyclophosphamide, and o-carboxyethyl-phosphoramide mustard. However, the pharmacokinetics and toxicities of these metabolites remain unclear [28]. Although its effectiveness has yet to be established, defibrotide is approved for the treatment of advanced sinusoidal obstruction syndrome after myeloablative therapy prior to hematopoietic cell transplantation, especially in the presence of pulmonary or renal abnormalities.

To prevent hepatotoxicity, natural antioxidants must be included in current chemotherapy treatments. The purpose of this work is to establish sesamin’s possible mechanism of defense against hepatotoxicity by cyclophosphamide through amelioration of oxidative stress and inflammatory cytokine apoptosis. According to a previous study, after the administration of cyclophosphamide, its toxic metabolites produce reactive oxygen species (ROS) that are harmful to the liver [29]. Hepatic fibrogenesis and liver injury are both significantly influenced by the production of ROS [30]. ROS are unstable, highly reactive molecules with a low molecular weight. Free radicals like the superoxide anion (O2*) and the hydroxyl radical (OH*) are produced by oxygen [31].

In this study, the administration of cyclophosphamide causes hepatocyte damage, as seen by increased levels of enzyme markers such as AST, ALT, ALP, and BIL. These indicators were found to be considerably higher in the group receiving cyclophosphamide only. Many hepatic abnormalities caused by free radicals are mediated by oxidative stress. Numerous studies have demonstrated the significance of free radicals that play a role in liver diseases caused by cyclophosphamide [32,33]. Animal studies over the past few decades have shown an association between free radical oxidative stress and cyclophosphamide-induced hepatotoxicity. Moreover, lipid peroxidation in hepatic cells caused by cyclophosphamide was accompanied by a significant decline in the levels of antioxidants (GSH) and enzymes (CAT and SOD) and a rise in MDA compared to the control. The damaging effects of cyclophosphamide due to free radicals and reactive oxygen species may be warded off by natural antioxidants [34,35,36]. Sesamin is a natural antioxidant that has unique antioxidant qualities that help to reduce MDA levels and improve GSH and antioxidant enzymes. Intriguingly, sesamin showed similar defensive effects against another chemotherapeutic. In a 2020 study by Ali B. H. and colleagues, sesamin’s protection was documented in counteracting cisplatin-induced nephrotoxicity in rats. This was achieved through the mitigation of inflammation, oxidative stress, and cellular injury [37].

Moreover, by investigating inflammatory cytokines, the critical function of pathogenesis in cyclophophamide-induced hepatotoxicity is illuminated. Inflammation and oxidative stress are positively correlated [38]. NF-kB activation, nuclear translocation, and upregulation are associated with inflammation, apoptosis, and fibrosis. It is well known that cyclophosphamide activates NF-kB p65, which prompts the transcription of interleukins (ILs), cytokines (TNFs), and fibrosis [39,40,41]. In response to ROS, NF-kB increases the production of fatty acid synthetase (FAS) ligands through transcriptional production, which causes apoptotic cell death.

Cytokines like TNFα and IL-1β are key mediators in the inflammatory process. TNFα is a cytokine that controls inflammation, apoptosis, and proliferation in the development of hepatotoxicity. Currently, there is ongoing discussion regarding TNF’s function following liver injury, despite the fact that it protects liver regeneration, too. A previous study demonstrated that the proteins’ activities that are involved in cell death are influenced by TNFα levels. It revealed that excessive amounts of TNFα have been demonstrated to worsen liver damage caused by lipopolysaccharides. The IL-1β converting enzyme, sometimes referred to as caspase-1, breaks down the precursor protein for IL-1β. The IL-1β precursor is additionally processed by other serine proteases. According to a previous study, the inflammatory cytokine IL-1β may make primary mouse hepatocytes more sensitive to the activation of caspases 3 and 7 by Fas ligand (FasL) [42,43].

In contrast to the normal control, the results showed a substantial rise in TNFα, IL-1β, and caspase-3 after cyclophosphamide treatment. Consistently, Alqahtani and Mahmoud (2016) showed that cyclophosphamide triggered hepatocyte death and observed a significant rise in the levels of caspase-3 protein and expression [44]. In addition, Caglayan et al. (2018) found that by elevating the level of cysteine aspartate-specific protease-3 caspase-3, cyclophosphamide (200 mg/kg b.w.) stimulated the autophagy and apoptotic pathways [45]. The present findings are consistent with those previously mentioned.

In sesame seeds and oil, sesamin serves as a prominent component, widely recognized for its cellular antioxidant, anti-inflammatory, and anti-apoptotic attributes. Its beneficial effects are produced by stopping the creation of inflammatory cytokines and ROS. Also, it suppresses the NF-kB pathway by blocking p65, which is detrimental to cell cycle equilibrium and plays a critical role in carcinogenesis. By raising antioxidant enzyme levels, it also aids in lowering ROS generation. In an evaluation contrasting the cyclophosphamide group, sesamin demonstrated enhanced remedial impacts on the liver’s redox status in the administered group. Notably, these effects were more pronounced at elevated sesamin doses compared to lower concentrations. However, the most substantial sesamin dose 20 mg/kg showed no significant deviation from the normal group.

The generation of inflammation, epithelial degradation, congestion, and vacuolization in hepatic tissue is another way that cyclophosphamide causes histological alterations. Many studies have documented instances of significant liver tissue enlargement and sinusoidal narrowing in rats treated with cyclophosphamide, which lends credence to this conclusion [46,47].

The aforementioned biochemical results were further supported by histological alterations and the ameliorative impact of sesamin in combination with cyclophosphamide treatment. The liver slices from rats treated with cyclophosphamide showed widespread degeneration, a notable filtration of inflammatory cells, sinusoidal obstruction, and necrosis in most of the hepatic parenchyma. Conversely, the tissue sections of the group that received sesamin alone did not exhibit any histological alterations. Sesamin treatment prevented the inflammation, necrosis, and sinusoidal obstruction architecture of liver tissue. It is evident that the higher dose of sesamin showed a greater protective effect than the lower dose, as shown in the injury score.

Although our study showed the beneficial effects of sesamin against cyclophosphamide-induced hepatotoxicity, further investigations such as protein synthesis, DNA alteration, serum γ glutamile transpeptidase, and serum ammonia should be conducted to gain a better understanding.

## 5. Conclusions

In conclusion, sesamin emerges as a potential antioxidant that mitigates the hepatotoxic effects induced by cyclophosphamide. This is achieved through the modulation of oxidative stress, inflammatory cytokines, apoptotic pathways, and histological changes. Consequently, sesamin could serve as a supplementary agent to ameliorate the detrimental impacts of cyclophosphamide on liver cells, potentially improving the efficacy of chemotherapy treatments.

## Figures and Tables

**Figure 1 biomedicines-11-03238-f001:**
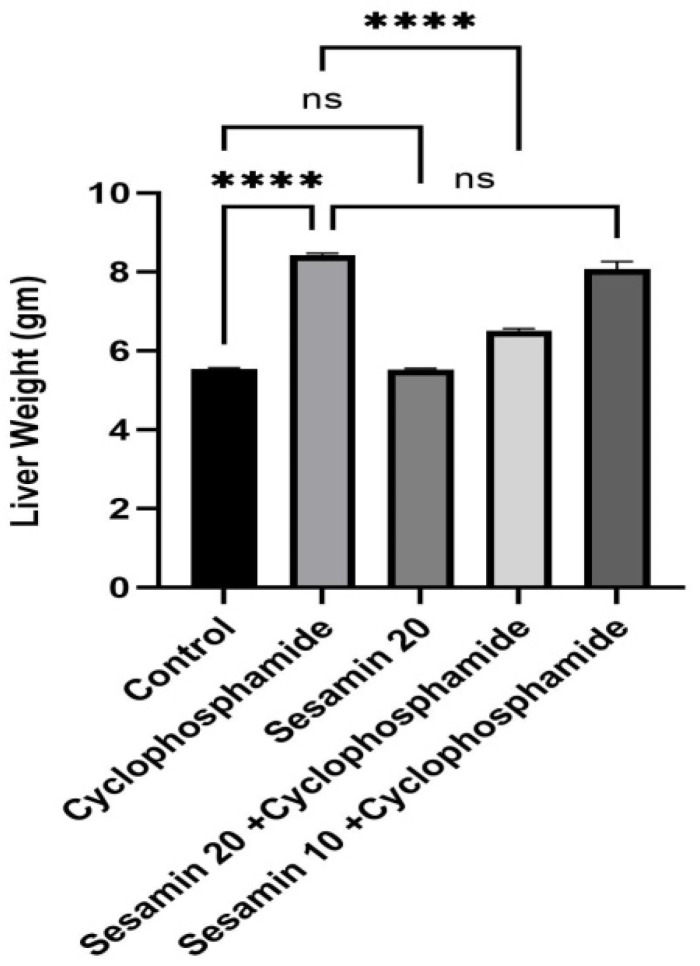
Effects of sesamin on liver weight against cyclophosphamide-induced hepatotoxicity in rats. Values are represented in Mean ± SEM (*n* = 6). **** *p* < 0.0001, ^ns^
*p* > 0.05 vs. control and cyclophosphamide.

**Figure 2 biomedicines-11-03238-f002:**
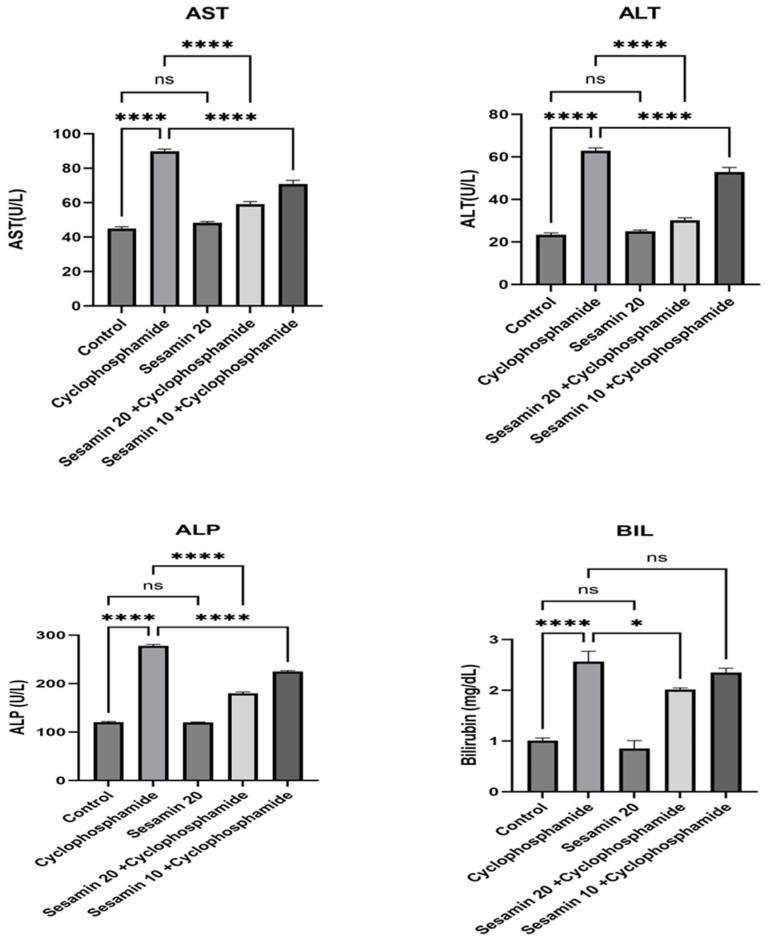
Effects of sesamin on hepatic enzymes (AST, ALT, ALP, and BIL) against cyclophosphamide-induced hepatotoxicity in rats. Values are represented in Mean ± SEM (*n* = 6). * *p* < 0.01, **** *p* < 0.00001, ^ns^
*p* > 0.05 vs. control and cyclophosphamide.

**Figure 3 biomedicines-11-03238-f003:**
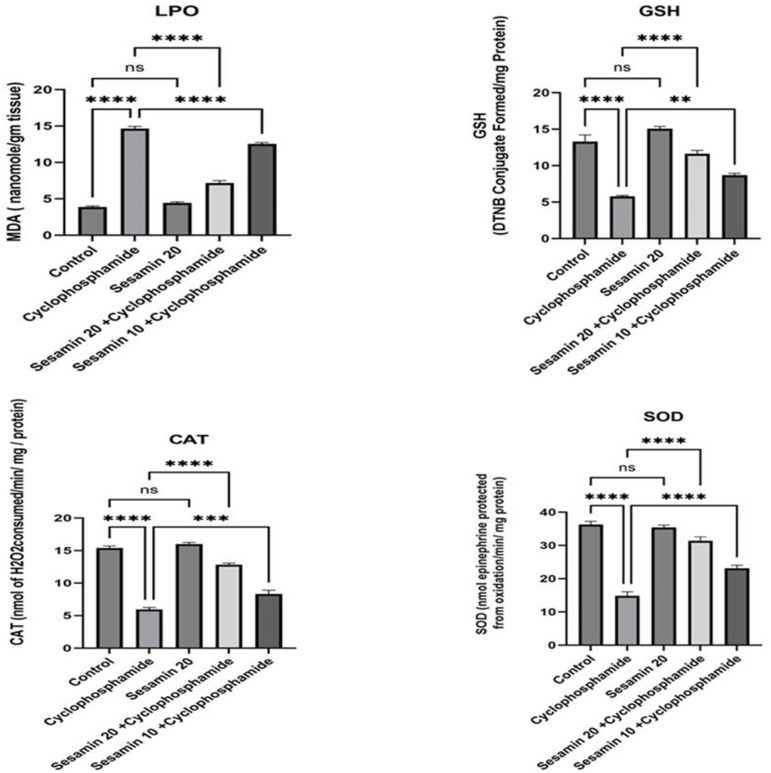
Sesamin’s effects on MDA, GSH, CAT, and SOD against cyclophosphamide-induced hepatotoxicity in rats. Values are represented in Mean ± SEM (*n* = 6). ** *p* < 0.001, *** *p* < 0.0001, **** *p* < 0.00001, ^ns^
*p* > 0.05 vs. control and cyclophosphamide.

**Figure 4 biomedicines-11-03238-f004:**
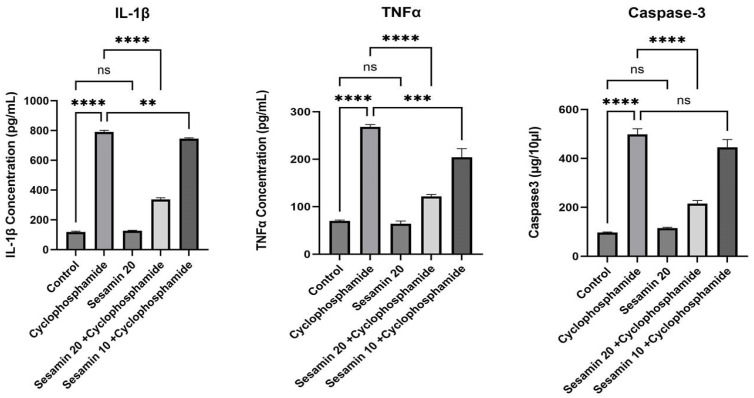
Effects of sesamin on inflammatory cytokines and apoptotic markers against cyclophosphamide in rats. Values are represented in Mean ± SEM (*n* = 6). ** *p* < 0.001, *** *p* < 0.0001, **** *p* < 0.00001, ^ns^
*p* > 0.05 vs. control and cyclophosphamide.

**Figure 5 biomedicines-11-03238-f005:**
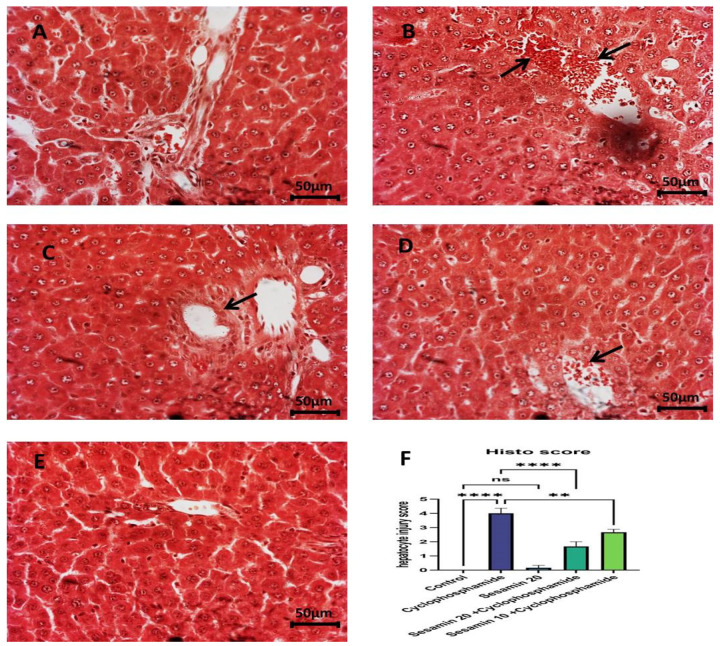
(**A**–**F**) Sesamin’s effects on liver histopathology against cyclophosphamide by using hamtoxylin and esosin (H&E) stain. (**A**) Control represents the normal histology without any changes with an injury score of 0. (**B**) Cyclophosphamide represents a clear abnormality such as inflammation, necrosis, and sinusoidal disturbances with an injury score of 4. (**C**,**D**) Cyclophosphamide + sesamin represents both doses (20 and 10 mg/kg) of sesamin treatment and recovery with injury scores of 2 and 3, respectively. (**E**) Sesamin 20 represents the alone treatment without any alteration with an injury score of 0. The effectiveness of sesamin against cyclophosphamide represented the hepatic injury score in figure (**F**). Cyclophosphamide vs. control is highly significant (**** *p* < 0.0001); sesamin 20 vs. control is not significant (^ns^
*p* > 0.05); and sesamin 20 and 10 + cyclophosphamide vs. cyclophosphamide is a significant restore (** *p* < 0.001, **** *p* < 0.0001). Arrows indicate the sinusoidal disturbances in the above pictures.

## Data Availability

Authors declare that data is available inside the article.

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
