# Peer review of "Sesamin’s Therapeutic Actions on Cyclophosphamide-Induced Hepatotoxicity, Molecular Mechanisms, and Histopathological Characteristics"

_biomedicines, 2023, doi:10.3390/biomedicines11123238_

Round 1

Reviewer 1 Report

Comments and Suggestions for Authors

1.      The description of experiment design is inconsistent between abstract and main manuscript.

The authors have to clarify if the oral sesamin at 10 and 20 mg/kg is pre-treated for seven days before giving a single dose of cyclophosphamide, or after receiving a single dose of cyclophosphamide.

2.      In Figure 4, the quality of the figures needs to be improved and the authors are suggested to compare the same liver regions between control and treatment groups. The scale bar for each figure is needed. The authors are suggested to label the portal or central vein, apoptosis, inflammatory cells and other injury observations in the figures, especially Figure 4B. The description of quantifying the hepatic injury score is needed in the figure legend.

Comments on the Quality of English Language

Proofread by a native English speaker is suggested.

Author Response

Response to The Rewviewer-1

Dear Sir

First, we want to thank you for taking the time to read and comment on our manuscript. Yours careful, constructive, and helpful reviews made it better. We have responded to yours comments as detailed below. We also proofread the entire manuscript meticulously and changes are highlighted in yellow color in the manuscript, while some word is deleted by marking red color.

Comments and Suggestions for Authors

  1. The description of experiment design is inconsistent between abstract and main manuscript.

Response: Assuming the reviewers are aware of the word count restrictions imposed by the abstract, I respectfully disagree with their suggestions. That's why we can't describe and repeat the same as manuscript in the abstract.

The authors have to clarify if the oral sesamin at 10 and 20 mg/kg is pre-treated for seven days before giving a single dose of cyclophosphamide, or after receiving a single dose of cyclophosphamide.

Response: It is made clear in the abstract that both amounts of sesamin were used as a pretreatment every day from day 1 to day 6, and that cyclophosphamide was given as a single dose on day 4. The reviewer asked this question because they may be missed to focus it in this paper.

  1. In Figure 4, the quality of the figures needs to be improved and the authors are suggested to compare the same liver regions between control and treatment groups. The scale bar for each figure is needed. The authors are suggested to label the portal or central vein, apoptosis, inflammatory cells and other injury observations in the figures, especially Figure 4B. The description of quantifying the hepatic injury score is needed in the figure legend.

Response: I appreciate the author’s comments and modified in the text and figure accordingly. I also added the scale bar and added the level for SOS. Figure legend also modified by added injury scores and highlighted in yellow colors.

Comments on the Quality of English Language

Proofread by a native English speaker is suggested.

Response: Thanks for your comments. I rectified all issues with the help of native English speaker as suggested by the reviewers.

Reviewer 2 Report

Comments and Suggestions for Authors

In this study, the authors developed cyclophosphamide induced liver toxicity in a rat model and then assessed the protective effect of Sesamin.

The authors previously developed nephrotoxicity using the same substance and tested Sesamin on the nephrological model.

Major points

1- The manuscript need extensive language profreeding. Even in the title include typos.

2- Liver toxicity in a rat model needs more markers to be assessed such as the effect of cyclophosphamide on : a) H2O2 generation  b) nitrite level      c) glutathione peroxidase.

Then the authors needs to assess the effect of Sesamin on these markers.

3) In the rat model, did the author measure the liver weight? other liver markers ?

4) What are the rationales for selecting IL-1 B and TNFa. Why not other inflammatory cytokines???? The same point for apoptosis marker.

Comments on the Quality of English Language

Extensive language editing is required

Author Response

Response To Reviewer-2 Comments

Dear Sir

First, we want to thank you for taking the time to read and comment on our manuscript. Yours careful, constructive, and helpful reviews made it better. We have responded to yours comments as detailed below. We also proofread the entire manuscript meticulously and changes are highlighted in yellow color in the manuscript, while some word is deleted by marking red color.

Comments and Suggestions for Authors

In this study, the authors developed cyclophosphamide induced liver toxicity in a rat model and then assessed the protective effect of Sesamin.

The authors previously developed nephrotoxicity using the same substance and tested Sesamin on the nephrological model.

Major points

1- The manuscript need extensive language profreeding. Even in the title include typos.

Response: I would appreciate if reviewers highlight the specified errors.

2- Liver toxicity in a rat model needs more markers to be assessed such as the effect of cyclophosphamide on : a) H2O2 generation  b) nitrite level      c) glutathione peroxidase.

Then the authors needs to assess the effect of Sesamin on these markers.

Response: I agree with reviewers, but I also examined lipid peroxidation LPO, which is strongly linked to H202 and Nitrite production because they generate free radicals and produce lipid peroxidation. I examined the H2O2-generating catalase. Due to limited project funds, we measured glulathione instead of glutathione peroxidase and other antioxidant enzymes.

3) In the rat model, did the author measure the liver weight? other liver markers ?

Response: Yes I had measure the liver weight but forget to include. Thanks for reminding and now I have included in the manuscript in Figrue.1.

4) What are the rationales for selecting IL-1 B and TNFa. Why not other inflammatory cytokines???? The same point for apoptosis marker.

Response: Inflammatory cytokines like interleukin-1 beta and tumor necrosis factor alpha (TNFa) mediate the activation and adhesion of circulating phagocytic cells and signal the end of the innate immune response. Persistent innate immune response activation or desregulation may contribute to the etiology of chronic inflammatory disorders. Due to a lack of funding, I was unable to conduct analyses on the other inflammatory cytokines as well as apoptotic markers.

Comments on the Quality of English Language

Extensive language editing is required    

Response: As per reviewers comments I have critically reviews with the help of English language expert and fixed all typo and grammar errors.

Reviewer 3 Report

Comments and Suggestions for Authors

The authors have presented their results on the hepatoprotective effects of sesamin on cyclophosphamide-induced hepatotoxicity in a rat model.  I have several questions and comments:

1. how were the doses of sesamin decided?  Were  they the  same doses used in other studies (e.g ref 37 line 299)? And if so, how were the optimal doses determined?

2. The 150mg/kg/i.p. dose of cyclophosphamide appears to be supratherapeutic compared to the human dose of 10-50mg/kg given IV every 2-10 days (or 1-5mg/kg orally). What is the mean time to onset for human hepatotoxicity (including SOS) and how does this latency compare to single dose cyclophosphamide in this model? 

3. line 347 states that sesamin "restored" normal liver architecture. But should this not be referred to as  "prevented" liver injury since you do not have paired liver histology to actually demonstrate recovery from acute injury?  In addition, has sesamin been shown to prevent or ameliorate sinusoidal obstruction syndrome in any models, since this is the most worrisome hepatic lesion that may occur? And were any measurements of nitric oxide (NO) made as this has been associated with develop of SOS in rats? 

4. the manuscript would be strengthened by the addition of other and more up to date references of other potentially cytoprotective compounds preventing or ameliorating cyclophosphamide injury in similar rat studies. The authors should then compare and contrast the findings of these other compounds with sesamin and provide a statement about their respective potential utility in this setting. 

5. why was alkaline phosphatase not measured ?

Comments on the Quality of English Language

few typos found - "investigated"

avoid use of slang expressions (e.g.  a 'slew" of ...)

Author Response

Response to Reviewer 3 Comments

Dear Sir

First, we want to thank you for taking the time to read and comment on our manuscript. Yours careful, constructive, and helpful reviews made it better. We have responded to yours comments as detailed below. We also proofread the entire manuscript meticulously and changes are highlighted in yellow color in the manuscript, While some word is deleted by marking red color.

Open Review

Quality of English Language

( ) I am not qualified to assess the quality of English in this paper
( ) English very difficult to understand/incomprehensible
( ) Extensive editing of English language required
( ) Moderate editing of English language required
(x) Minor editing of English language required
( ) English language fine. No issues detected

Yes

Can be improved

Must be improved

Not applicable

Does the introduction provide sufficient background and include all relevant references?

(x)

( )

( )

( )

Are all the cited references relevant to the research?

( )

(x)

( )

( )

Is the research design appropriate?

(x)

( )

( )

( )

Are the methods adequately described?

( )

(x)

( )

( )

Are the results clearly presented?

(x)

( )

( )

( )

Are the conclusions supported by the results?

(x)

( )

( )

( )

Comments and Suggestions for Authors

The authors have presented their results on the hepatoprotective effects of sesamin on cyclophosphamide-induced hepatotoxicity in a rat model.  I have several questions and comments:

  1. how were the doses of sesamin decided?  Were they the  same doses used in other studies (e.g ref 37 line 299)? And if so, how were the optimal doses determined?

Response: Thanks for your comments. The sesamin doses was decided on the basis of previous study reference number 18, and 19

  1. The 150mg/kg/i.p. dose of cyclophosphamide appears to be supratherapeutic compared to the human dose of 10-50mg/kg given IV every 2-10 days (or 1-5mg/kg orally). What is the mean time to onset for human hepatotoxicity (including SOS) and how does this latency compare to single dose cyclophosphamide in this model?

Response: Thanks for your comments and I agree with you that 150mg/kg/ip dose of CP appears to be supratherapeutic compared to human dose. This doses was selected on the basis of previous reports that include 150-200 mg/kg body weight ip as a  single dose for the induction of hepto and renal diseases. 

  1. line 347 states that sesamin "restored" normal liver architecture. But should this not be referred to as  "prevented" liver injury since you do not have paired liver histology to actually demonstrate recovery from acute injury?  In addition, has sesamin been shown to prevent or ameliorate sinusoidal obstruction syndrome in any models, since this is the most worrisome hepatic lesion that may occur? And were any measurements of nitric oxide (NO) made as this has been associated with develop of SOS in rats?

Response: Thanks for your valuable comment and suggestion and I rectified the term “ restored”. Also marked in histological slides of sinusoidal obstruction and improvement by sesamin treatment. I did not measured the nitric oxide due to funds limitations. 

  1. the manuscript would be strengthened by the addition of other and more up to date references of other potentially cytoprotective compounds preventing or ameliorating cyclophosphamide injury in similar rat studies. The authors should then compare and contrast the findings of these other compounds with sesamin and provide a statement about their respective potential utility in this setting. 

Response: I appreciate the reviewer suggestion and accordingly I have modified the text with highlight and updated the references.

  1. why was alkaline phosphatase not measured ?

 Response: I received ALP kits late from supplier but now I have examined the ALP and added in the manuscript as Figure 1. 

Comments on the Quality of English Language

few typos found - "investigated" avoid use of slang expressions (e.g.  a 'slew" of ...)

Response: Thanks for your remarks and I removed typo error and also removed “slew”

Submission Date

12 October 2023

Date of this review

23 Oct 2023 20:03:01

Round 2

Reviewer 1 Report

Comments and Suggestions for Authors

The author did not address some of the questions.

I am confused by the statement that:

in abstract, the authors stated that the experimental treatment group were "two groups pretreated with oral sesamin (10 and 20 mg/kg daily for six days) followed by an i.p. injection 20 of cyclophosphamide on day four, and a group receiving a consistent high oral dose of sesamin throughout the study".

In the experiment section, the authors wrote that "The next two groups were given intraperitoneal (150 mg/kg) cyclophosphamide on day four and oral sesamin at 10 and 20 mg/kg (named Sesamin 10+ cyclophosphamide and Sesamin 20+ cyclophosphamide, respectively) for seven days [19]".

These description are not clear, at least to me, about the experiment design. If the authors meant "It is made clear in the abstract that both amounts of sesamin were used as a pretreatment every day from day 1 to day 6, and that cyclophosphamide was given as a single dose on day 4." Please state it clear as it is in both abract and manuscript.

Comments on the Quality of English Language

The English need to be improved, like the above mentioned experiment design is not clear.

Author Response

Round 2 Responses to Reviewer 1

Respected Sir

We appreciate you reading and providing feedback on our manuscript. It was improved by your thoughtful, informative, and useful criticisms. As you can see below, we have addressed your comments. In addition, we did a thorough proofreading of the entire work, including references and highlighted.

Open Review

Quality of English Language

( ) I am not qualified to assess the quality of English in this paper
( ) English very difficult to understand/incomprehensible
( ) Extensive editing of English language required
(x) Moderate editing of English language required
( ) Minor editing of English language required
( ) English language fine. No issues detected

Yes

Can be improved

Must be improved

Not applicable

Does the introduction provide sufficient background and include all relevant references?

( )

( )

(x)

( )

Are all the cited references relevant to the research?

( )

( )

(x)

( )

Is the research design appropriate?

( )

( )

(x)

( )

Are the methods adequately described?

( )

( )

(x)

( )

Are the results clearly presented?

( )

( )

(x)

( )

Are the conclusions supported by the results?

( )

( )

(x)

( )

Comments and Suggestions for Authors

The author did not address some of the questions.

I am confused by the statement that:

in abstract, the authors stated that the experimental treatment group were "two groups pretreated with oral sesamin (10 and 20 mg/kg daily for six days) followed by an i.p. injection 20 of cyclophosphamide on day four, and a group receiving a consistent high oral dose of sesamin throughout the study".

Response: Yes, I do agree that the experimental group took two doses of oral sesamin pretreatment with 10 and 20 mg/kg body weight daily for six days. However, I disagree that I administered a 20 i.p. injection of cyclophosphamide on day four, as one of the esteemed reviewers pointed out that I failed to findout it anywhere in the abstract. I have modified the abstract in simple language and highlighted as per reviewer suggestion.

In the experiment section, the authors wrote that "The next two groups were given intraperitoneal (150 mg/kg) cyclophosphamide on day four and oral sesamin at 10 and 20 mg/kg (named Sesamin 10+ cyclophosphamide and Sesamin 20+ cyclophosphamide, respectively) for seven days [19]".

These description are not clear, at least to me, about the experiment design. If the authors meant "It is made clear in the abstract that both amounts of sesamin were used as a pretreatment every day from day 1 to day 6, and that cyclophosphamide was given as a single dose on day 4." Please state it clear as it is in both abract and manuscript.

Response: Thanks for your valuable comments and I have changed the language in simple way and keep both place abstract and manuscript similar.

Comments on the Quality of English Language

The English need to be improved, like the above mentioned experiment design is not clear.

Response: As per your suggestion modified the experimental design and highlighted.

Submission Date

12 October 2023

Date of this review

13 Nov 2023 05:32:08

Reviewer 2 Report

Comments and Suggestions for Authors

No further comments

Comments on the Quality of English Language

Moderate language editing

Author Response

Round 2 Responses to Reviewer 2

Respected Sir

Your time in reviewing and commenting on our manuscript is greatly appreciated. Your constructive critique was insightful, instructive, and ultimately helpful in making it better. Your insightful comments have been much appreciated.

Open Review

Quality of English Language

( ) I am not qualified to assess the quality of English in this paper
( ) English very difficult to understand/incomprehensible
( ) Extensive editing of English language required
(x) Moderate editing of English language required
( ) Minor editing of English language required
( ) English language fine. No issues detected

Yes

Can be improved

Must be improved

Not applicable

Does the introduction provide sufficient background and include all relevant references?

(x)

( )

( )

( )

Are all the cited references relevant to the research?

(x)

( )

( )

( )

Is the research design appropriate?

(x)

( )

( )

( )

Are the methods adequately described?

(x)

( )

( )

( )

Are the results clearly presented?

(x)

( )

( )

( )

Are the conclusions supported by the results?

(x)

( )

( )

( )

Comments and Suggestions for Authors

No further comments

Comments on the Quality of English Language

Moderate language editing

Response: Yes again I go through the entire manuscript thoroughly and changed the complicated sentences to simple sentences as per your suggestion.

Submission Date

12 October 2023

Date of this review

11 Nov 2023 01:54:03

Round 3

Reviewer 1 Report

Comments and Suggestions for Authors

The author addressed the questions and it is ready to be published.